# Seedling Establishment and Yield Performance of Dry Direct-Seeded Rice after Wheat Straw Returning Coupled with Early Nitrogen Application

**Jinyu Tian** [1], **Shaoping Li** [1], **Zhipeng Xing** [1], **Shuang Cheng** [1], **Qiuyuan Liu** [2], **Lei Zhou** [1], **Ping Liao** [1], **Yajie Hu** [1], **Baowei Guo** [1], **Haiyan Wei** [1] and **Hongcheng Zhang** [1,*]

[1] Jiangsu Key Laboratory of Crop Cultivation and Physiology/Innovation Center of Rice Cultivation Technology in Yangtze Valley, Ministry of Agriculture/Co-Innovation Center for Modern Production Technology of Grain Crops, Yangzhou University, Yangzhou 225009, China; dx120190081@yzu.edu.cn (J.T.); lsp2226992705@163.com (S.L.); zpxing@yzu.edu.cn (Z.X.); cs1692135738@163.com (S.C.); zlzzll95@126.com (L.Z.); p.liao@yzu.edu.cn (P.L.); huyajie@yzu.edu.cn (Y.H.); gbwyx@126.com (B.G.); wei_haiyan@163.com (H.W.)

[2] Agricultural College, Xinyang Agriculture and Forestry University, Xinyang 464000, China; liuqy@xyafu.edu.cn

\* Correspondence: hczhang@yzu.edu.cn

**Abstract:** Dry direct-seeded rice sown by multifunctional seeders (MS-DDSR) has received increased attention for its high efficiency. Wheat straw returning is widely used as an important agricultural practice because it is the simplest and quickest approach to dispose of wheat straw and also improve soil quality. The study determined whether MS-DDSR after wheat straw returning could obtain a high yield and whether early nitrogen (N) application could compensate for the negative effects caused by returned wheat straw. Field experiments were performed in a split-plot design. Main plots were comprised without wheat straw returning (S0) and wheat straw returning (S1). Split plots consisted of three plots with early N application treatment: 65 (N1), 95 (N2), and 125 (N3) kg N ha$^{-1}$ at 0 and 20 days after sowing. S1 reduced yield, N uptake, and biomass accumulation in MS-DDSR compared to S0 because S1 negatively affected the seedling roots growth, seedling establishment, and tillering capacity of MS-DDSR. The positive interaction between wheat straw returning and early N on yield, biomass accumulation, and N uptake was likely related to the positive interaction on spikelet number per panicle, total spikelet number, and biomass accumulation after the stem elongation stage. These findings demonstrate that wheat straw returning led to poor seedling establishment and yield loss for MS-DDSR, but these negative effects could be compensated for by an appropriate increase in early N application, based on the locally recommended N application protocols.

**Keywords:** seedling establishment; yield performance; wheat straw returning; early nitrogen application; dry direct-seeded rice

## 1. Introduction

Dry direct seeding, an ancient method of rice establishment, is based on sowing dry rice seeds directly in the main field without saturated and puddled soils [1]. Historically, dry direct-seeded rice (DDSR) was mostly concentrated in rainfed areas of Asia, Africa, Latin America, and the Caribbean [2,3]. More recently, a looming water crisis [4] and escalating labor costs [5] have driven the gradual expansion of the cultivation area in well-irrigated areas such as the middle and lower reaches of the Yangtze River in China and the upper Ganges River delta in India [6–8]. In addition, the development of multifunctional seeders that perform synchronous rotary tillage and sowing [9,10] and the improvement of weed management technology [11,12] provide opportunities to improve the efficiency and increase the yield of DDSR. Dry direct-seeded rice sown by multifunctional seeders (MS-DDSR) was eventually developed.

Rice–wheat rotation is an important agricultural production system in countries in south Asia, such as China and India [13,14]. Because of the short turnaround time after combine harvesting of wheat, wheat straw management is a challenging problem in the rice–wheat rotation system [15]. Open field burning of wheat straw is the common practice [16,17], but it has led to serious impacts on air quality, health, and climate [18–20]. Thus, wheat straw returning by rotary tillage is widely used as an important agricultural measure because it is the simplest and quickest approach to dispose of wheat straw and meet the time requirement in the rice–wheat rotation system. However, rice seedlings exhibit stress after wheat straw returning. For example, wheat straw returning leads to the accumulation of microbial allelochemicals, such as organic acids and toxic soil reductants, during wheat straw decomposition, restricting root development of rice seedlings [21–24]. Wheat straw returning may increase the yield of transplanted rice [25–28] because it can ameliorate soil structure and improve soil organic matter content, and its additional nutrients and positive effects can alleviate its negative effects on rice growth [23,29,30]. Nonetheless, relative to transplanted rice seedlings that already possess some level of stress resistance when transplanting from the seedling raising land to the main field, MS-DDSR is always under stress from seed germination to seedling growth, leading to poor seedling establishment. The differences between MS-DDSR and transplanted rice make it unlikely that MS-DDSR sown after wheat straw returning would also increase yield. To our knowledge, some studies have been conducted under conditions of wet direct-seeding [31–34], pot planting [21,35], and manual direct-seeding (i.e., broadcasting) [36–38], but few studies have been conducted on the effects of MS-DDSR after wheat straw returning. Seedling establishment and yield performance of MS-DDSR after wheat straw returning require further evaluation, confirmation, and data support in the rice–wheat rotation system.

Nitrogen (N) is an essential nutrient for rice growth [39,40]. However, MS-DDSR caused higher N loss than transplanted rice because of increased ammonia volatilization [41] and surface runoff [42] in the early growth stage. The higher C:N ratio (~80:1) of wheat straw also results in high N immobilization and mineralization during wheat straw decomposition, especially within 30 days after wheat straw returning [43], thereby causing N deficiency in rice seedlings in the early growth stage [44,45]. Moreover, early N application of basal and tillering fertilizers plays a decisive role in the root development, seedling establishment, tillering occurrence, biomass accumulation, and yield performance of rice [46]. Therefore, early N application is particularly important for MS-DDSR after wheat straw returning. Wang [31] reported that alternate wetting and drying and furrow irrigation could alleviate the negative effects of wheat straw returning by increasing root and shoot biomass, root oxidation activity, and harvest index. Yang [36] reported that yield reduction in direct-seeded rice after wheat straw returning could be mitigated by appropriate water management and improving soil phosphorus (P) availability. Previous studies on alleviating the negative effects on rice after wheat straw returning mainly focused on water management and reducing wheat straw returning. However, whether early N application can compensate for the negative effects on seedling establishment and yield performance of MS-DDSR after wheat straw returning is unclear.

The objectives of this study were (1) to evaluate the seedling establishment and yield performance of MS-DDSR after wheat straw returning, and (2) to clarify the mechanisms of how early N application can alleviate or compensate the negative effects caused by wheat straw returning. The findings of this study provide theoretical and practical support for high-yield cultivation and regulatory approaches of MS-DDSR after wheat straw returning.

## 2. Materials and Methods

### 2.1. Experimental Site and Weather Conditions

Field experiments were performed in 2019 and 2020 on a typical rice-wheat rotation system at the research farm of Yangzhou University, Jiangsu, China (32°61′ N, 120°12′ E). The field soil was a sandy loam with a viscous texture and 30.4 g kg$^{-1}$ organic matter, 1.91 g kg$^{-1}$ total N, 31.6 mg kg$^{-1}$ available P, and 154 mg kg$^{-1}$ available potassium (K).

The amount of wheat straw returning was approximately 6.51 t ha$^{-1}$ biomass on average in both years. The sowing date was 11 June and the harvest date was October 20 in both years. Meteorological data for daily mean temperature, sunshine hours, and precipitation during the rice growing season in 2019 and 2020 were collected at a weather station near the experimental site (Figure 1).

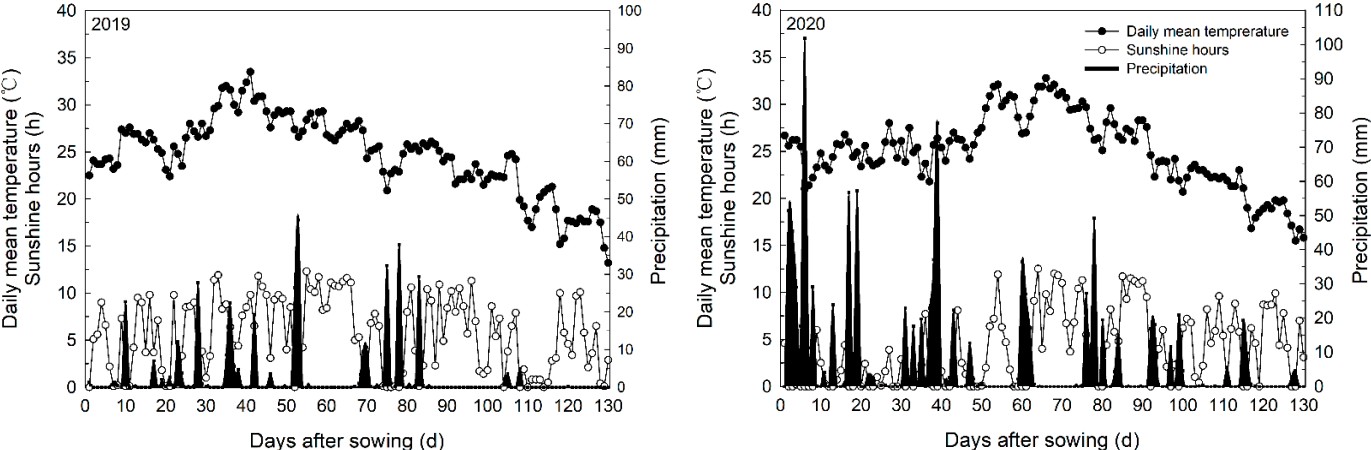

**Figure 1.** Daily mean temperature, sunshine hours, and precipitation during the rice growth season in 2019 and 2020.

## 2.2. Experimental Design and Treatment Modes

The experiment was performed using a split-plot design. Main plots were the wheat straw returning treatment methods: control without wheat straw returning (S0) and wheat straw returning (S1). Split plots were the three early N application modes (urea with 45.6% N content): 65 kg N ha$^{-1}$ (N1), 95 kg N ha$^{-1}$ (N2, locally recommended N application), and 125 kg N ha$^{-1}$ (N3) at 0 and 20 days after sowing (DAS; the information of nitrogen application are shown in Table 1). In all plots, P fertilizer of calcium superphosphate (12% $P_2O_5$ content; 135 kg P ha$^{-1}$) was applied at pre-sowing (0 DAS), and K fertilizer of potassium chloride (60% $K_2O$ content; 135 kg K ha$^{-1}$) was applied at pre-sowing and again at the panicle initiation stage (50 DAS). Treatments were arranged in the same experimental plots in both years. The experiments were replicated four times with an area of 36 m$^2$ in each plot. The japonica rice variety Nanjing-9108 was selected in this experiment.

**Table 1.** Information on nitrogen application in this experiment.

| Treatment | | 0 DAS [1] (kg ha$^{-1}$) | 20 DAS (kg ha$^{-1}$) | 50 DAS (kg ha$^{-1}$) | 70 DAS (kg ha$^{-1}$) |
|---|---|---|---|---|---|
| Without wheat straw returning | N1 | 65 | 65 | 40 | 40 |
| | N2 | 95 | 95 | 40 | 40 |
| | N3 | 125 | 125 | 40 | 40 |
| Wheat straw returning | N1 | 65 | 65 | 40 | 40 |
| | N2 | 95 | 95 | 40 | 40 |
| | N3 | 125 | 125 | 40 | 40 |

[1] DAS: days after sowing; 0 DAS: sowing stage; 20 DAS: early tillering stage; 50 DAS: panicle initiation stage; 70 DAS: initiation of spikelet differentiation stage.

In both years, wheat was harvested by a combine and wheat straw was returned to the field. Wheat straw in the main plots (where wheat straw was not returned) was manually removed, and fertilizer was applied to all plots. Next, rice seeds (dry seeds without germination) were seeded with a multifunctional seeder that performs synchronous rotary tillage and sowing. The depth of rotary tillage was 10−15 cm, and the seedling rate was 130 kg ha$^{-1}$. After sowing, the plots with the different treatment modes were surrounded

by bunds (width 40 cm; height 30 cm), which were covered with plastic film. Until the three-leaf stage, the seedlings in each plot were thinned to 260 m$^{-2}$.

### 2.3. Crop Cultivation

After sowing, wet irrigation management was used during the seedling period to ensure the growth and development of rice seedlings. Thereafter, moistening conditions were maintained until the five-leaf stage. The field was flooded after the five-leaf stage, and the water level was maintained at 2–3 cm until the middle tillering stage. Water was then drained for 7–10 days to control unproductive tillers. After the stem elongation stage, an alternate wetting and drying irrigation management regime was used until one week before the final harvest. In the alternate wetting and drying irrigation management regime, fields were not irrigated until the soil water potential reached −15 kPa (soil moisture content 0.170 g g$^{-1}$) at 15–20 cm soil depth. The soil water potential was monitored at 15–20 cm soil depth using a tensiometer with a 5 cm sensor (Institute of Soil Science, Chinese Academy of Sciences, Nanjing, Jiangsu, China). Four tensiometers were installed in each plot, and readings were recorded at 11:00 each day. When the soil water potential reached the threshold, all plots were flooded with 3 cm of water. The irrigation water level was monitored with a LXSG-50 flow meter (Shanghai Water Meter Manufacturing Factory, Shanghai, China) installed on the irrigation pipelines.

### 2.4. Sampling and Measurements

#### 2.4.1. Seedling Characteristics

At 35 DAS, 20 consecutive seedlings were randomly selected from each plot to measure their morphological characteristics, such as height, leaf area, and base diameter. Leaf area was measured with a LI-3100 leaf area meter (LI-COR, Lincoln, NE, USA). The base diameter was measured with Vernier calipers. Seedlings were oven-dried at 105 °C for 30 min and then at 80 °C in bags to a constant weight, and the seedling weight was measured. N uptake per seedling was calculated by multiplying the N concentration per seedling (%) by the biomass per plant, and N concentration was determined by the semi-micro-Kjeldahl procedure.

#### 2.4.2. Root Morphological Characteristics

At 35 DAS, a sampling core was used for root sampling, in which a 25 cm × 16 cm × 20 cm cube of soil was removed from each plot. The cube was placed in a nylon net, and the soil was removed with running water. To determine the root morphological characteristics, such as length, average diameter, surface area, and volume, roots were scanned with a flatbed image scanner (Epson Expression 1600 scanner, Torrance, CA, USA), and the images were analyzed using WinRhizo software (v5.0, Regent Instruments, Quebec City, QC, Canada). Roots were oven-dried at 105 °C for 30 min and then at 80 °C in bags to a constant weight, and the root weight was measured.

#### 2.4.3. Dynamic Tillering Model

In the two years, three adjacent rows (1 m length) in each plot were measured at five-day intervals from the three-leaf stage until the tiller number diminished.

Tiller number (Y) was fit with a logistic equation:

$$Y = \frac{Y_m}{1 + exp(A - Bx)},$$

where $Y_m$ is the maximum tiller number, x is DAS, and *A* and *B* are rate-controlling parameters [47]. The fitting was performed using Origin 9 (Origin Lab Corporation, Northampton, MA, USA).

#### 2.4.4. Oxidation–Reduction Potential

In situ soil oxidation–reduction potential (Eh) was determined by potentiometry (FJA-6; Institute of Soil Science, Nanjing, China) at 0, 10, 20, 30, and 40 DAS, respectively. Each measurement was taken at 08:00. The depth of measurement was 10 cm below the soil surface, and five replicates were performed for each plot.

#### 2.4.5. Biomass

Biomass was determined at the stem elongation, heading, and maturity stages. Samples were collected from three adjacent rows (50 cm × 75 cm) in each plot, oven-dried at 105 °C for 30 min and then at 80 °C in bags to a constant weight, and the weight was measured.

#### 2.4.6. Yield and Yield Components

At the maturity stage, three adjacent rows (1 m length) were sampled randomly from each plot for measurement of panicle traits and yield components, including spikelet number per panicle, filled grain percentage, and 1000-grain weight. Three adjacent rows (5 m length) were sampled randomly from each plot to determine the panicle number per $m^2$. Total spikelet number per $m^2$ was calculated as panicle number per $m^2$ multiplied by spikelet number per panicle. Yield was determined from the harvest area of 8 $m^2$ in each plot and adjusted to the standard moisture content of 0.14 g $H_2O$ $g^{-1}$.

#### 2.5. Statistical Analysis and Formula Calculation

Data were analyzed using IBM SPSS Statistics 22 (SPSS, Chicago, IL, USA), and treatment means were compared by the least significant difference test. Graphical representations of the data were produced using Origin 9 (Origin Lab Corporation, Northampton, MA, USA), Microsoft Excel 2019 (Microsoft Corporation, Washington, DC, USA), and SigmaPlot 12.0 (Systat Software, Inc., San Jose, CA, USA).

### 3. Results

#### 3.1. Yield and Yield Components

Wheat straw returning and early N application significantly affected yield and N uptake at maturity of MS-DDSR (Table 2). Compared with S0, S1 had 7.2% lower yield and 9.4% lower N uptake in the N1 treatment mode, and 4.6% lower yield and 7.1% lower N uptake in the N2 treatment mode. Compared with the N1 treatment mode, the N3 treatment mode had 7.2% higher yield and 18.4% higher N uptake, and the N2 treatment mode had 4.7% higher yield and 9.7% higher N uptake. In terms of yield components, S1 had a significantly increased spikelet number per panicle, but a significantly decreased panicle number compared with S0. The panicle number was 3.9% and 5.4% higher for the N2 and N3 treatment modes compared with the N1 treatment mode. Wheat straw × N application positively affected yield, N uptake, spikelet number per panicle, and total spikelet number per $m^2$ (Table 2 and Figure 2).

The correlation analysis over the two years (Figure 3) showed a significant positive correlation between yield and panicle number, total spikelet number per $m^2$, biomass, and N uptake at maturity. No significant correlation was observed between yield and filled grain percentage and 1000-grain weight.

**Table 2.** Average yield, N uptake, and yield components at maturity as affected by wheat straw returning (S0 or S1; *n* = 18), nitrogen (N1, N2, or N3; *n* = 12), and year (2019 or 2020; *n* = 18).

| | Yield (t ha$^{-1}$) | N Uptake (kg ha$^{-1}$) | Panicles (m$^{-2}$) | Spikelets (Panicle$^{-1}$) | Total Spikelets (m$^{-2}$) | Filled Grain (%) | Grain Weight (mg) |
|---|---|---|---|---|---|---|---|
| Year (Y) [1] | | | | | | | |
| 2019 | 10.0 | 181 | 403 | 103 | 41392 | 94.7 | 26.6 |
| 2020 | 9.86 ** | 180 | 391 ** | 104 | 40655 * | 95.9 * | 26.5 |
| Wheat straw (S) [2] | | | | | | | |
| S0 | 10.2 | 186 | 407 | 102 | 41383 | 95.2 | 26.5 |
| S1 | 9.79 *** | 175 *** | 388 *** | 105 *** | 40664 * | 95.4 | 26.7 |
| Nitrogen (N) [3] | | | | | | | |
| N1 | 9.60 | 165 | 385 | 102 | 39232 | 95.7 | 26.5 |
| N2 | 10.1 | 181 | 400 | 104 | 41553 | 95.3 | 26.6 |
| N3 | 10.3 *** | 196 *** | 406 *** | 104 ** | 42286 *** | 95.0 | 26.7 |
| F-values [4] | | | | | | | |
| S × N | 11.1 *** | 3.96 * | 5.42 * | 7.95 ** | 15.5 *** | NS | NS |
| S × Y | NS | NS | NS | NS | NS | NS | NS |
| N × Y | NS | NS | NS | NS | NS | NS | NS |

Significant treatment effects within a main category are indicated by * ($0.01 < p \leq 0.05$), ** ($0.001 < p \leq 0.01$), and *** ($p \leq 0.001$). [1] Values were averaged across wheat straw and nitrogen. [2] Values were averaged across nitrogen and year. [3] Values were averaged across wheat straw and year. [4] F-values are provided for interactions. There were no significant three-way interactions.

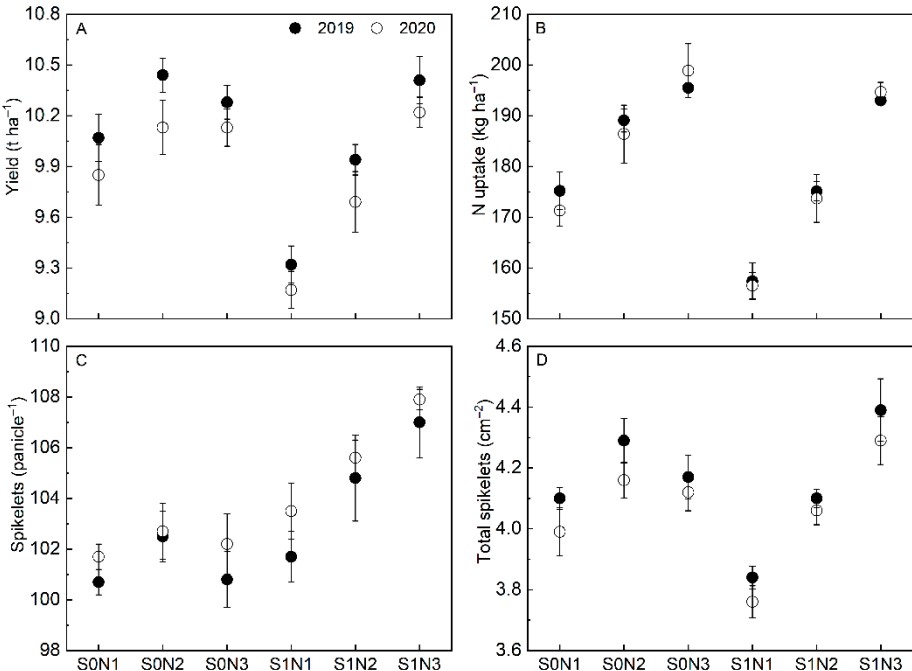

**Figure 2.** Yield (**A**); N uptake (**B**); spikelets per panicle (**C**); and total spikelets number (**D**) for all treatment combinations in 2019 and 2020. S0 and S1 represent treatment without wheat straw returning and treatment with wheat straw returning, respectively. N1, N2, and N3 represent early N management of 65 kg N ha$^{-1}$, 95 kg N ha$^{-1}$, and 125 kg N ha$^{-1}$, respectively. The information on these treatments was shown in Table 1. Error bars show standard error of replicates (*n* = 3).

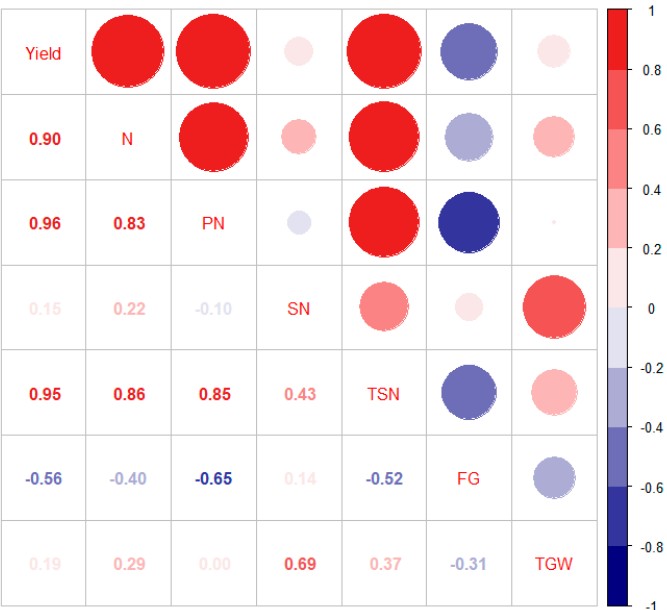

**Figure 3.** Correlation analysis among yield components, N uptake, and biomass at maturity ($R_{0.05} = 0.576$ and $R_{0.01} = 0.707$). N: N uptake at maturity; PN: panicles number; SN: spikelets number per panicle; TSN: total spikelet number per $m^2$; FG: filled grain percentage; GW: grain weight.

### 3.2. Biomass Accumulation

Wheat straw returning significantly affected biomass accumulation, mainly affecting biomass accumulation from the sowing to the stem elongation stage of MS-DDSR (Table 3). Compared with S0, S1 had 7.8% lower biomass accumulation at maturity in the N1 treatment mode and 5.7% lower biomass accumulation at maturity in the N2 treatment mode. Biomass accumulation from the sowing to the stem elongation stage was 8.1% lower for S1 than for S0.

**Table 3.** Biomass accumulation as affected by wheat straw returning (S0 or S1; $n = 18$), nitrogen (N1, N2, or N3; $n = 12$), and year (2019 or 2020; $n = 18$).

| | Biomass (MA) (t ha$^{-1}$) | Biomass (SO-SE) (t ha$^{-1}$) | Biomass (SE-HD) (t ha$^{-1}$) | Biomass (HD-MA) (t ha$^{-1}$) |
|---|---|---|---|---|
| Year (Y) [1] | | | | |
| 2019 | 17.3 | 4.12 | 6.77 | 6.45 |
| 2020 | 16.7 *** | 3.97 ** | 6.86 | 5.92 *** |
| Wheat straw (S) [2] | | | | |
| S0 | 17.4 | 4.21 | 6.87 | 6.35 |
| S1 | 16.7 *** | 3.87 *** | 6.75 | 6.03 ** |
| Nitrogen (N) [3] | | | | |
| N1 | 16.3 | 3.57 | 6.73 | 5.99 |
| N2 | 17.2 | 4.01 | 6.94 | 6.16 |
| N3 | 17.7 *** | 4.47 *** | 6.77 | 6.41 ** |
| F-values [4] | | | | |
| S × N | 11.0 *** | NS | 7.06 ** | 3.91 * |
| S × Y | NS | NS | NS | NS |
| N × Y | NS | NS | NS | NS |

Significant treatment effects within a main category are indicated by * ($0.01 < p \leq 0.05$), ** ($0.001 < p \leq 0.01$), and *** ($p \leq 0.001$). Biomass (MA), Biomass (SO-SE), Biomass (SE-HD), and Biomass (HD-MA) represent biomass accumulation at maturity, biomass accumulation from sowing to stem elongation stage, biomass accumulation from stem elongation to heading stage, and biomass accumulation from heading to maturity stage, respectively. [1] Values were averaged across wheat straw and nitrogen. [2] Values were averaged across nitrogen and year. [3] Values were averaged across wheat straw and year. [4] F-values are provided for interactions. There were no significant three-way interactions.

Early N application significantly affected biomass accumulation at all growth stages. Compared with the N1 treatment mode, the N3 treatment mode had 8.4% higher biomass accumulation and the N2 treatment mode had 5.5% higher biomass accumulation at maturity. Wheat straw × N application positively affected biomass accumulation at maturity, from the stem elongation to the heading stage, and from the heading stage to the maturity stage (Table 3 and Figure 4).

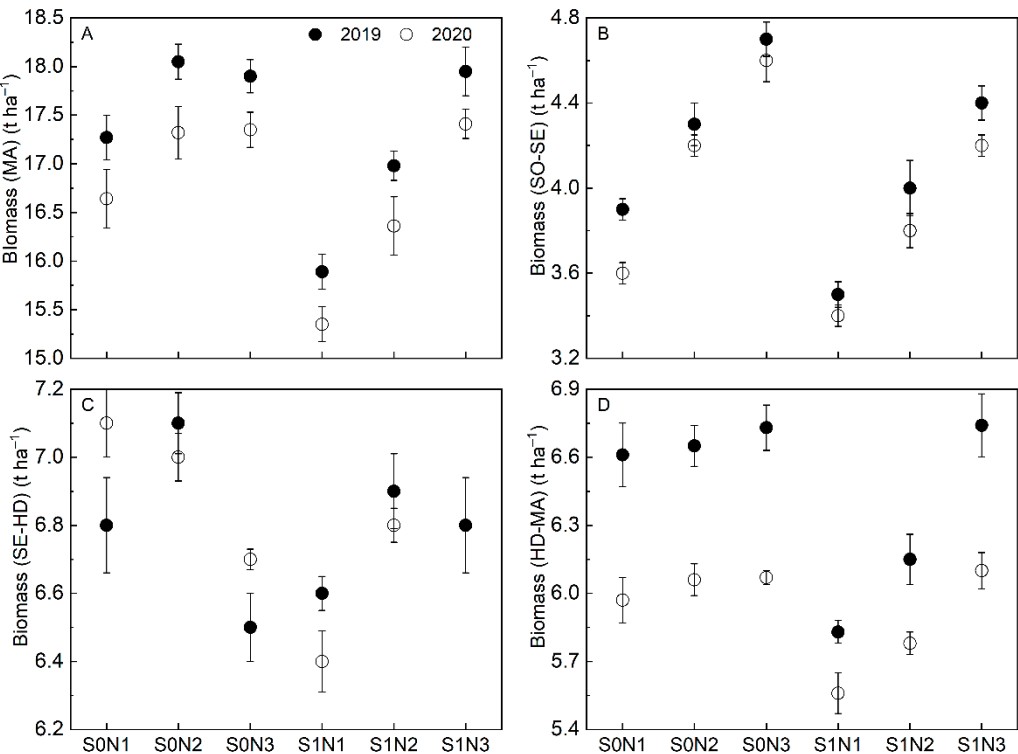

**Figure 4.** Biomass accumulation at MA (**A**); SO-SE (**B**); SE-HD (**C**); and HD-MA (**D**) for all treatment combinations in 2019 and 2020. SO: sowing stage; SE: stem elongation stage; HD: heading stage; MA: maturity stage. S0 and S1 represent treatment without wheat straw returning and treatment with wheat straw returning, respectively. N1, N2, and N3 represent early N management of 65 kg N ha$^{-1}$, 95 kg N ha$^{-1}$, and 125 kg N ha$^{-1}$, respectively. The information of these treatments was shown in Table 1. Error bars show standard error of replicates (*n* = 3).

### 3.3. Tillering Dynamics

From the dynamic tillering model across the two years (Figure 5), the estimated maximum tiller number was 4.8–5.2% lower for S1 than for S0. Compared with S0, S1 had 5.4–7.3% lower maximum tiller number in the N1 treatment mode and 3.8–7.7% lower maximum tiller number in the N2 treatment mode. Increasing early N application could compensate for the negative effects of wheat straw returning and increase the maximum tiller number.

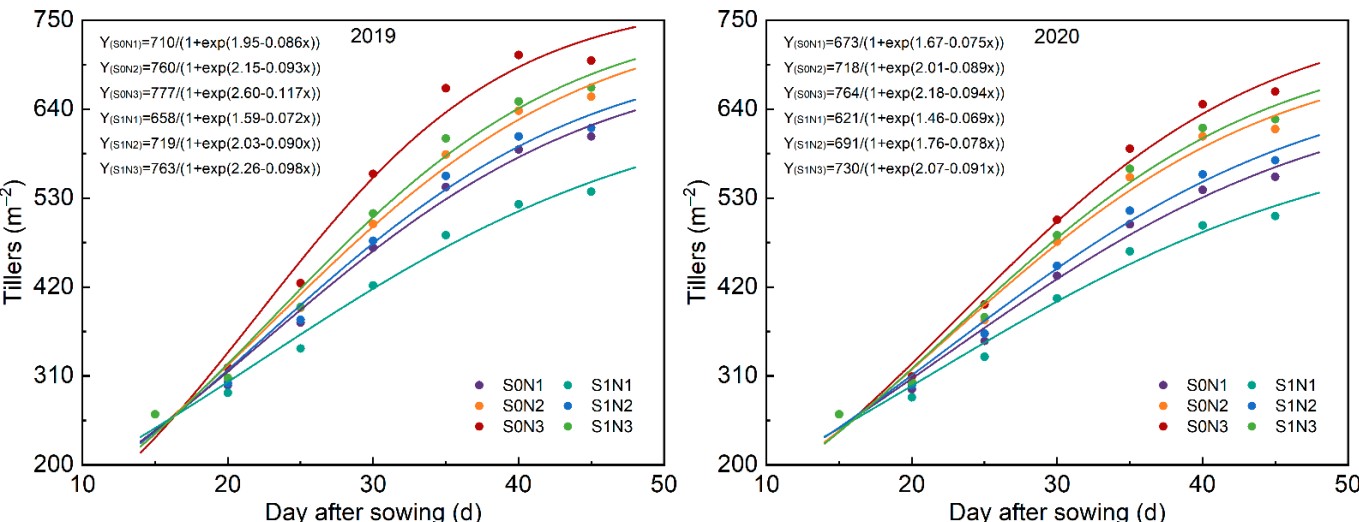

**Figure 5.** Tillering dynamics under wheat straw returning coupled with early N management in 2019 and 2020. S0 and S1 represent treatment without wheat straw returning and treatment with wheat straw returning, respectively. N1, N2, and N3 represent early N management of 65 kg N ha$^{-1}$, 95 kg N ha$^{-1}$, and 125 kg N ha$^{-1}$, respectively. The information on these treatments was shown in Table 1.

### 3.4. Seedling Characteristics

Wheat straw returning negatively affected seedling establishment (Table 4). This was reflected in the seedlings by the reduction in height by 4.6%, weight by 5.8%, leaf area by 6.1%, base diameter by 7.1%, and N uptake by 7.3%. In the N1 treatment mode, S1 decreased seedling height by 5.9%, weight by 7.5%, leaf area by 8.3%, base diameter by 10.5%, and N uptake by 9.0%, compared with S0. Similarly, in the N2 treatment mode, S1 decreased seedling height by 5.4%, weight by 5.7%, leaf area by 6.9%, base diameter by 6.1%, and N uptake by 6.7%, compared with S0. Early N application positively affected seedling establishment. The interactions between wheat straw returning, nitrogen, and year were not significant for seedling morphological characteristics and N uptake at 35 days after sowing.

**Table 4.** Seedling morphological characteristics and N uptake at 35 days after sowing as affected by wheat straw returning (S0 or S1; *n* = 18), nitrogen (N1, N2, or N3; *n* = 12), and year (2019 or 2020; *n* = 18).

|  | Height (cm) | Weight (mg) | Leaf Area (cm$^2$) | Base Diameter (mm) | N Uptake (mg) |
|---|---|---|---|---|---|
| Year (Y) [1] |  |  |  |  |  |
| 2019 | 22.8 | 87.8 | 7.77 | 5.01 | 2.75 |
| 2020 | 21.4 ***[4] | 81.8 *** | 6.49 *** | 4.23 *** | 2.53 *** |
| Wheat straw (S) [2] |  |  |  |  |  |
| S0 | 22.6 | 87.4 | 7.35 | 4.79 | 2.74 |
| S1 | 21.6 *** | 82.3 ** | 6.90 *** | 4.45 ** | 2.54 *** |
| Nitrogen (N) [3] |  |  |  |  |  |
| N1 | 20.5 | 80.6 | 6.51 | 4.13 | 2.43 |
| N2 | 22.0 | 84.8 | 7.07 | 4.61 | 2.65 |
| N3 | 23.8 *** | 89.0 *** | 7.80 *** | 5.12 *** | 2.84 *** |

[1] Values were averaged across wheat straw and nitrogen. [2] Values were averaged across nitrogen and year. [3] Values were averaged across wheat straw and year. [4] Significant treatment effects within a main category are indicated by ** (0.001 < *p* ≤ 0.01), and *** (*p* ≤ 0.001).

### 3.5. Root Morphological Characteristics

Wheat straw returning and early N application significantly affected the root morphological characteristics of MS-DDSR (Table 5). Compared with S0, S1 led to restricted root growth, which was reflected in the seedling roots by the reduction in weight by 6.0%, length by 6.5%, average diameter by 4.2%, surface area by 5.3%, and volume by 5.1%. Increasing early N application could increase seedling root weight by 7.1%, length by 8.1%, average diameter by 5.7%, surface area by 7.6%, and volume by 6.7% in the N1 treatment mode. Similarly, in the N2 treatment mode, increasing early N application could increase seedling root weight by 6.1%, length by 7.3%, average diameter by 4.8%, surface area by 5.1%, and volume by 5.2%. The interactions between wheat straw returning, nitrogen, and year were not significant for root morphological characteristics at 35 days after sowing.

**Table 5.** Root morphological characteristics at 35 days after sowing as affected by wheat straw returning (S0 or S1; *n* = 18), nitrogen (N1, N2, or N3; *n* = 12), and year (2019 or 2020; *n* = 18).

| | Root Weight (mg) | Root Length (cm) | Average Diameter (mm) | Root Surface Area (mm$^2$) | Root Volume (mm$^3$) |
|---|---|---|---|---|---|
| Year (Y) [1] | | | | | |
| 2019 | 10.4 | 90.6 | 0.40 | 1109 | 100 |
| 2020 | 9.90 *** | 87.4 ** | 0.40 | 1080 * | 96.0 ** |
| Wheat straw (S) [2] | | | | | |
| S0 | 10.5 | 91.9 | 0.41 | 1124 | 100 |
| S1 | 9.82 * | 86.0 *** | 0.39 * | 1065 *** | 95.0 ** |
| Nitrogen (N) [3] | | | | | |
| N1 | 9.46 | 80.1 | 0.37 | 1042 | 92.0 |
| N2 | 10.2 | 88.8 | 0.40 | 1095 | 98.0 |
| N3 | 10.8 *** | 98.1 *** | 0.43 *** | 1146 *** | 104 *** |

Significant treatment effects within a main category are indicated by * ($0.01 < p \le 0.05$), ** ($0.001 < p \le 0.01$), and *** ($p \le 0.001$). [1] Values were averaged across wheat straw and nitrogen. [2] Values were averaged across nitrogen and year. [3] Values were averaged across wheat straw and year.

### 3.6. Oxidation–Reduction Potential

S1 could increase soil Eh within 15 DAS (Figure 6), but Eh gradually decreased after about 15 DAS compared with S0. The Eh value was 128–179% lower for S1 than for S0 at 40 DAS. Increasing early N application reduced the Eh value within 40 DAS.

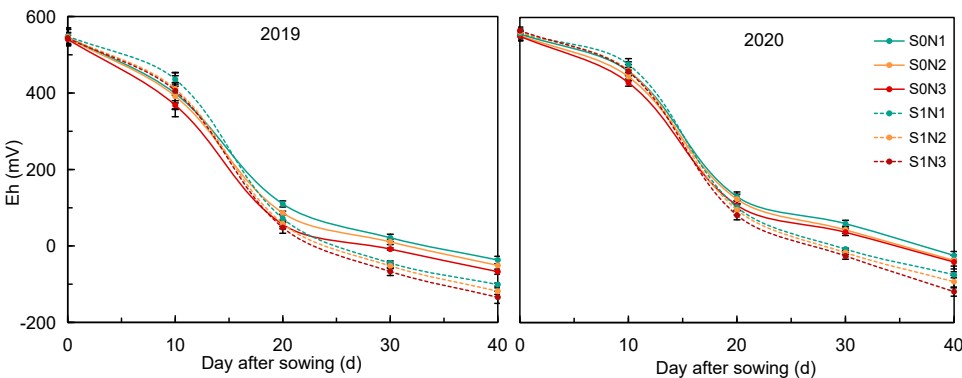

**Figure 6.** The oxidation–reduction potential (Eh) during 40 day after sowing under wheat straw returning coupled with early N management in 2019 and 2020. S0 and S1 represent treatment without wheat straw returning and treatment with wheat straw returning, respectively. N1, N2, and N3 represent early N management of 65 kg N ha$^{-1}$, 95 kg N ha$^{-1}$, and 125 kg N ha$^{-1}$, respectively. The information on these treatments was shown in Table 1. Error bars show standard error of replicates (*n* = 3).

## 4. Discussion

Dry direct-seeded rice sown by multifunctional seeders that perform synchronous rotary tillage and sowing has received increased attention. Wheat straw returning is widely used as an important agricultural practice because it is the simplest and quickest approach to dispose of wheat straw and also improve soil quality in the rice–wheat rotation system [28,48]. This study showed that wheat straw returning negatively affected seedling establishment, which was reflected by decreased height, weight, leaf area, and base diameter at 35 DAS, and lower biomass accumulation from the sowing to the stem elongation stage for MS-DDSR seedlings. These results were similar to the results of Du [49], in which wheat straw returning reduced the tiller number, plant height, dry weight, and NPK concentrations of rice seedlings. Enhanced N fixation by soil microbes during wheat straw decomposition decreases N absorbed by plants [50,51]. Azam [52] reported that the uptake of soil N by plants decreased with wheat straw returning. The present study also showed that wheat straw returning decreased N uptake by seedlings (Table 4). Therefore, N deficiency in seedlings contributed to poor seedling establishment of MS-DDSR after wheat straw returning.

The root system, as a bridge for information and material exchange between plant and soil, performs the functions of absorbing and transporting nutrients, as well as synthesizing and storing nutrients; it also plays a crucial role in seedling establishment. Tanaka [53] reported that the decline in crop N uptake caused by straw returning originated from N uptake inhibition. Du [49] reported that rice roots after wheat straw returning showed low growth, poor vigor, and insufficient nutrient uptake capacity. In the present study, wheat straw returning negatively affected the seedling root system, which was reflected by lower weight, length, average diameter, surface area, and volume of roots of MS-DDSR seedlings (Table 5). There are two possible reasons for the restricted seedling root system after wheat straw returning. First, the formation of strongly reducing conditions after wheat straw returning restricted the growth of the seedling roots. Soil reductants, such as ferroporphyrin sulfite and reducible organic acids, accumulate in strongly reducing conditions after wheat straw returning in paddy soils [21]. In addition, excessive ferrous iron concentrations lead to rice malnutrition by antagonistic effects on the uptake of essential nutrients [54]. Sulfite can be reduced to $H_2S$ under strongly reducing conditions, resulting in damage to seedling roots [21]. Soil Eh is an important indicator of the degree of oxidation or reduction in soil [55]. The soil Eh value after wheat straw returning decreased with the advancement of straw returning and soil flooding (Figure 6). This result was consistent with the findings of Yang [36] that straw returning under continuous flooding might intensify the soil reducing strength, as determined by a causal relationship analysis using structural equation modeling. Second, microbial allelochemicals, such as phenolic acids, accumulate during the decomposition of returned wheat straw [22,56]. These allelochemicals may negatively affect the physiological and biochemical processes of rice seedlings, such as inhibiting cell division and elongation, destroying cell membrane integrity, and changing enzyme activity [57,58]. Therefore, poor seedling establishment was caused by wheat straw returning that originated from the restricted root system of the MS-DDSR seedlings.

Wheat straw returning negatively affected the yield performance of MS-DDSR (Table 2). This result was consistent with the findings of [21] and Yang [36] that showed lower yields of DDSR after wheat straw returning. In terms of yield components, wheat straw returning tended to increase the spikelets per panicle, filled grain percentage, and 1000-grain weight, but it led to a decrease in the panicle number without wheat straw returning. These results were consistent with previous studies [28,32]. The increase in spikelets per panicle, filled grain percentage, and 1000-grain weight could be related to the fact that nutrient release after wheat straw decomposition was beneficial to rice grain filling. The correlation analysis indicated that the panicle number was the key factor to determine the yield and total spikelet number per $m^2$ (Figure 3). Therefore, the insufficient panicle number was the main reason for the reduced yield of MS-DDSR after wheat straw returning. The maximum tiller number is one of the primary factors determining the panicle number. In this study,

the estimated maximum tiller number based on the dynamic tillering model was greater in the field without wheat straw returning than in the field with wheat straw returning (Figure 5). Gao [21] reported that rice responded strongly to straw returning in the early growth stage, and that the tiller number (at about six weeks after sowing) was significantly reduced by straw returning. Wheat straw returning reduces the incidence of low tiller position or early emerging tillers [59]. Therefore, limited seedling tillering capacity after wheat straw returning may be the main reason for the decreased panicle number of MS-DDSR. In addition, limited seedling tillering may be because of poor seedling establishment after wheat straw returning, as seedling quality can reflect seedling tillering capacity to a certain extent [60].

Given the current harvest index of rice, improving biomass accumulation, especially after the heading stage [61], may be key to further improvement of rice yield [62,63]. The biomass accumulates of rice increased after wheat straw returning because additional organic matter and nutrients brought about by wheat straw returning may promote growth in the middle and late growth stages of rice [64–66]. However, this study showed that biomass accumulation and yield after wheat straw returning were reduced for MS-DDSR. Wheat straw returning is beneficial for increasing biomass and yield, mainly realized by transplanting rice, in previous research [25,28]. Thus, additional organic matter and nutrients brought about by wheat straw returning could not alleviate poor seedling establishment for MS-DDSR. The negative effects on MS-DDSR could also not be compensated for by the positive effects after wheat straw returning. There are two possible reasons for the difference between our results (i.e., MS-DDSR) and those of previous research (i.e., transplanting rice). First, seedling establishment of MS-DDSR was poorer compared with transplanting rice. When transplanting from the seedling raising area to the main field with wheat straw returned soil, the transplanted rice seedlings had at least four leaves, indicating that they had reached the stress resistance stage and acquired some stress resistance. However, MS-DDSR was always under stress caused by wheat straw returning. Second, after wet tillage, wheat straw was fully mixed in the 0–25 cm soil layer in the main field for transplanting rice, which reduced the negative effects of wheat straw. However, the rotary tillage depth of MS-DDSR was only 0–15 cm, which caused most wheat straw to assemble on the soil surface, increasing its toxic effects.

Early N application positively affected seedling establishment and yield performance. Wheat straw × N application had no effect on biomass accumulation before the stem elongation stage, N uptake, and seedling growth at 35 DAS, indicating that the negative effects caused by wheat straw returning could be mitigated by the positive effects of increased early N application. There are three possible reasons. First, increasing early N application could enhance seedling root development and compensate for the restricted seedling root system brought about by wheat straw returning. The N supply level affects root structure and physiological characteristics [67]. N deficiency severely affects the growth and development of the root system [68]. Increasing N application can promote cytokinin biosynthesis and acropetal transport [69], thereby enhancing seedling root development. Furthermore, low to medium N availability enhances root growth and branching to optimally utilize the macronutrient, whereas high levels of N availability may limit root growth [70]. In the present study, the seedling root system showed a promoting effect at the highest N level. It also showed that N was still deficient for MS-DDSR seedlings after wheat straw returning in the early growth stage (i.e., increased N was still in the range of root promotion). The possible reasons were that (1) returned wheat straw with a higher C:N ratio led to a decrease in soil available N in the early stage of decomposition [43,71], and (2) MS-DDSR easily increased ammonia volatilization [41,72] and N runoff losses [42] during the early growth stage because of the very shallow ponding water and large soil porosity. Second, increasing early N application could improve seedling N uptake and growth and increase stress tolerance of MS-DDSR seedlings after wheat straw returning. Third, increasing N application can decrease the accumulation of organic acids [73] and reduce the negative effects of wheat straw returning, thereby increasing its beneficial effects [64,74].

Increasing early N application led to a decrease in Eh (Figure 6). The possible reasons were that (1) N application motivates soil N-cycle microbial communities [75], and the promotion of their metabolic activities increases the consumption of soil-dissolved oxygen [76], which further induces a decrease in soil Eh [77], and (2) N application promotes the decomposition of wheat straw, thereby producing more reductants leading to a lower soil Eh. Although increasing early N application could not alleviate the reducing condition in the soil, it could compensate for the negative effects caused by wheat straw returning by promoting root development and increasing N uptake to improve seedling establishment.

The positive interaction between wheat straw returning and early N application on yield, biomass accumulation, and N uptake at maturity was likely related to the positive interaction between wheat straw returning and early N application on spikelet number per panicle, total spikelet number per m$^2$, and biomass accumulation after the stem elongation stage. The positive interaction may be because of several reasons. First, the initially immobilized N in the early growth stage was released in the middle to late growth stage. Second, increasing early N application can promote wheat straw decomposition, optimize the release of organic matter and nutrients of wheat straw, and provide a sufficient supply for biomass accumulation after the stem elongation stage [27]. Third, increasing early N application could compensate for the N loss caused by wheat straw returning, promote N absorption, seedling establishment (Tables 4 and 5), and tillering capacity (Figure 5), thereby providing a strong foundation for biomass accumulation after the stem elongation stage (Table 3). Fourth, increasing early N application increased the base diameter of seedlings after wheat straw returning, which was beneficial for resistance to rice lodging and enhanced light transmission in the rice population, thus favoring a positive interaction effect on yield.

Seedling establishment and yield performance were better in 2019 than in 2020. This difference could be explained by the weather differences between the two years (Figure 1). The duration of sunshine hours was 45.6% higher in 2019 than in 2020 during the rice growing season. In addition, the precipitation was 183.5% higher in 2020 than in 2019 during the rice growing season, especially during the early growth stage, which resulted in N loss.

## 5. Conclusions

Wheat straw returning negatively affected the seedling root system, seedling establishment, tillering capacity, biomass accumulation, and yield performance of MS-DDSR. Although increasing early N application could compensate for these negative effects, it is not advisable because additional N input would increase production costs, and excessive N input would lead to yield losses, reduced N utilization, and serious environmental consequences. Approaches such as improving the quality of tillage and increasing the seedling density of MS-DDSR after wheat straw returning will be the focus of future research.

**Author Contributions:** Field experiment and sampling analysis, J.T., S.L., S.C., Q.L. and L.Z.; writing—original draft preparation, J.T.; writing—review and editing, Z.X., P.L., Y.H., B.G. and H.W.; funding acquisition, J.T., Z.X., H.W. and H.Z. All authors have read and agreed to the published version of the manuscript.

**Funding:** This research was funded by the Jiangsu Agriculture Science and Technology Innovation Fund, grant number CX(20)1012; the Jiangsu Demonstration Project of Modern Agricultural Machinery Equipment and Technology, grant number NJ2020-58; the Jiangsu Technical System of Rice Industry, grant number JATS[2020]432; the National Natural Science Foundation of China, grant number 31801293; the National Technical System of Rice Industry, grant number CARS–01–27; and the Yangzhou University Scientific Research and Innovation Program, grant number XKYCX20_022.

**Institutional Review Board Statement:** Not applicable.

**Informed Consent Statement:** Not applicable.

**Data Availability Statement:** Not applicable.

**Acknowledgments:** We are grateful for grants from the Jiangsu Agriculture Science and Technology Innovation Fund, the Jiangsu Demonstration Project of Modern Agricultural Machinery Equipment and Technology, the Jiangsu Technical System of Rice Industry, the National Natural Science Foundation of China, the National Technical System of Rice Industry, and the Yangzhou University Scientific Research and Innovation Program. We would like to thank the editor and the reviewers for their useful feedback that improved this paper.

**Conflicts of Interest:** The authors declare no conflict of interest.

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
