# Peer review of "Seedling Establishment and Yield Performance of Dry Direct-Seeded Rice after Wheat Straw Returning Coupled with Early Nitrogen Application"

_agriculture, doi:10.3390/agriculture12040565_

Round 1

Reviewer 1 Report

It is well known that wheat straw returning has negative effect of the subsequent crop growth. Thus, it is difficult to find scientific novelty. Although the authors emphasize the originality in the study using direct seeding rice, the authors should compare direct seeding and transplant in a single study for this purpose. Nevertheless, this manuscript provides valuable field data conducted in two years with a number of related parameters, Thus if the following are properly amended, this manuscript will be better.

Line 2: deeded? -> seeded?

Line 101-102: 30.36, 31.57, 153.8, too many significant numbers. Please correct appropriately.

Line 115: Please describe how topdressing was done for urea and KCl.

Line 120-121: Please clarify whether 36 m2 is a total area or a area for each replicate. The current expression is confusable.

Line 122-125: Please clarify the sequence of 1) wheat straw return, 2) fertilizer application, 3) seeding, 4) rotary tillage.

Line 129; Please describe how the authors thinned to 260 m-2.

Line 140: please describe how irrigation management was done, including frequency and intensity.

Line 188: Please describe how many plants were samples. In addition, it is better to add information on the size in which plants were sampled. 50cm length is not enough. 50 cm x ?? cm.

Line 223: Please consider the significant numbers more carefully. 180.88 kg N is too many.

Line 252: same comment for Table 2.

Line 292: same comment for Table 2

Line 311: same comment for Table 2.

Line 319: “improve” is a very confusable term.

Line 323; For either treatment S0 or S1, the solid lines should be changed to dotted lines. In figure for 2020, A0 should be S0 and A1 should be S1.

Line 389-390: why MS-DDSR easily increased ammonia volatilization is not clear. Please add an explanation for the mechanism.

Line 400: “could not alleviate the adverse soil environment”, this is correct?

Line 405: the sections 4.1 and 4.2 should be combined. The effects of wheat straw retuning on plant growth should be discussed as a whole. Some discussion parts are overlapping both in 4.1 and 4.2.

Author Response

Dear Reviewer,

Thank you very much for your valuable comments and suggestions on our manuscript. Those comments are all valuable and very helpful for revising and improving our paper, as well as the important guiding significance to our research. We have studied comments carefully and have made the correction which we hope meet with approval. The main corrections in the manuscript and the responses to the reviewer’s comments are as flowing (in red color):

Comments and Suggestions for Authors:

It is well known that wheat straw returning has negative effect of the subsequent crop growth. Thus, it is difficult to find scientific novelty. Although the authors emphasize the originality in the study using direct seeding rice, the authors should compare direct seeding and transplant in a single study for this purpose. Nevertheless, this manuscript provides valuable field data conducted in two years with a number of related parameters, Thus if the following are properly amended, this manuscript will be better.

Point 1: Line 2: deeded? -> seeded?

Response 1: Thank you very much for pointing this out. We have made the revision in the manuscript. (Line 3)

Point 2: Lines 101-102: 30.36, 31.57, 153.8, too many significant numbers. Please correct appropriately.

Response 2: We have made the revision in the manuscript. (Lines 99-100)

Point 3: Line 115: Please describe how topdressing was done for urea and KCl.

Response 3: In this experiment, the urea was applied a total of four times. The first application was made before sowing and then rototilled into the soil by the multifunctional seeders. The topdressing of urea was applied by hand to the surface of the soil in each plot at 20, 50, and 70 days after sowing. The KCl was applied a total of two times, and the first time was consistent with the application of urea. The topdressing of KCl was applied by hand to the surface of the soil in each plot at 50 days after sowing.

Point 4: Lines 120-121: Please clarify whether 36 m2 is a total area or a area for each replicate. The current expression is confusable.

Response 4: Thank you very much for pointing this out. The 36 m2 is a area for each plot. We have made the revision in the manuscript. (Lines 118-119)

Point 5: Line 122-125: Please clarify the sequence of 1) wheat straw return, 2) fertilizer application, 3) seeding, 4) rotary tillage.

Response 5: The multifunctional seeders used in this experiment was a rotary tiller in front and a seeder at the back. Therefore these sequences were 1) wheat straw return, 2) fertilizer application, 3) rotary tillage, 4) seeding.

Point 6: Line 129; Please describe how the authors thinned to 260 m-2.

Response 6: The row spacing of the multifunctional seeder used in this experiment is 25 cm, so 260 m−2 is projected to 65 seedlings per meter per row. A scale was used to ensure that there were 13 seedlings evenly spaced within 20 cm of each row in the field.

Point 7: Line 140: please describe how irrigation management was done, including frequency and intensity.

Response 7: In this experiment, the soil should be kept moist from sowing to emergence, but not saturated to avoid seed rotting. Apply irrigation (in case of no rain) after sowing in dry soil, followed by saturating the field at the three-leaf stage is essential. This practice will not only ensure good rooting and seedling establishment but also enhance germinate higher number of seeds. Thereafter, moistening conditions were maintained until the five-leaf stage. The field was flooded after the five-leaf stage, and the water level was maintained at 2−3 cm until the middle tillering stage. Water was then drained for 7−10 days to control unproductive tillers. After the stem elongation stage, fields were not irrigated until the soil water potential reached −15 kPa (soil moisture content 0.170 g g−1) at 15−20 cm soil depth. The soil water potential was monitored at 15−20 cm soil depth using a tensiometer with a 5-cm sensor. When the soil water potential reached the threshold, all plots were flooded with 3 cm of water. Irrigation management was stopped until one week before the final harvest.

Point 8: Line 188: Please describe how many plants were samples. In addition, it is better to add information on the size in which plants were sampled. 50cm length is not enough. 50 cm x ?? cm.

Response 8: Thank you very much for pointing this out. The size of the sample for this experiment is 50 cm x 75 cm, and approximately 90−100 seedlings. We have made the revision in the manuscript. (Line 188)

Point 9: Line 223: Please consider the significant numbers more carefully. 180.88 kg N is too many.

Response 9: We have made the revision in the manuscript. (Line 223)

Point 10: Line 252: same comment for Table 2.

Response 10: We have made the revision in the manuscript. (Line 252)

Point 11: Line 292: same comment for Table 2

Response 11: We have made the revision in the manuscript. (Line 292)

Point 12: Line 311: same comment for Table 2.

Response 12: We have made the revision in the manuscript. (Line 311)

Point 13: Line 319: “improve” is a very confusable term.

Response 13: Thank you very much for pointing this out. We have made the revision in the manuscript. (Line 318)

Point 14: Line 323; For either treatment S0 or S1, the solid lines should be changed to dotted lines. In figure for 2020, A0 should be S0 and A1 should be S1.

Response 14: We really thank you for pointing this out and also express our sincere respect to your rigor, enthusiasm, and responsibility. We have made the revision in the manuscript. (Line 321)

Point 15: Lines 389-390: why MS-DDSR easily increased ammonia volatilization is not clear. Please add an explanation for the mechanism.

Response 15: The MS-DDSR is moist management without a water layer in the early stage, as well as its soil porosity is large, thus increasing ammonia volatilization. This modification has been added in the manuscript. (Lines 429-431)

Point 16: Lines 400: “could not alleviate the adverse soil environment”, this is correct?

Response 16: Thank you very much for pointing this out. We have revised this statement in the manuscript in order to make it more reasonable. (Lines 441-443)

Point 17: Line 405: the sections 4.1 and 4.2 should be combined. The effects of wheat straw retuning on plant growth should be discussed as a whole. Some discussion parts are overlapping both in 4.1 and 4.2.

Response 17: Thank you very much for this important comment. Considering your constructive suggestion, we have combined the sections 4.1 and 4.2 of discussion in the manuscript. (Lines 328-465)

We appreciate for your warm work earnestly and hope that the correction will meet with approval.

Special thanks to you for your good comments.

Best regards

Sincerely yours,

Jinyu Tian and co-authors

Jiangsu Key Laboratory of Crop Cultivation and Physiology / Innovation Center of Rice Cultivation Technology in Yangtze Valley, Ministry of Agriculture / Co-Innovation Center for Modern Production Technology of Grain Crops, Yangzhou University, Yangzhou 225009, China

Reviewer 2 Report

In this study, authors studied the establishment of the seedlings and yield performance of dry direct-seeded rice after wheat straw returning together with early nitrogen application. In my opinion, the study is planned nicely. The data looks robust and interesting. The manuscript is also written well. However, I have few comments that can be considered for improving the quality of manuscript.

  1. Title should be catchy and easy to understand to attract the readers. The current title also has typo error (dry direct- deeded rice). Please modify it.
  2. In the abstract please provide some background information of the proposed study.
  3. In all table legends, please mention the number of samples used for calculating the values.
  4. There are several typographical, and errors throughout the manuscript.

Author Response

Dear Reviewer,

Thank you very much for your valuable comments and suggestions on our manuscript. Those comments are all valuable and very helpful for revising and improving our paper, as well as the important guiding significance to our research. We have studied comments carefully and have made the correction which we hope meet with approval. The main corrections in the manuscript and the responses to the reviewer’s comments are as flowing (in red color):

Comments and Suggestions for Authors:

In this study, authors studied the establishment of the seedlings and yield performance of dry direct-seeded rice after wheat straw returning together with early nitrogen application. In my opinion, the study is planned nicely. The data looks robust and interesting. The manuscript is also written well. However, I have few comments that can be considered for improving the quality of manuscript.

Point 1: Title should be catchy and easy to understand to attract the readers. The current title also has typo error (dry direct- deeded rice). Please modify it.

Response 1: Thank you very much for pointing this out. We have made the revision in the manuscript. (Line 3)

Point 2: In the abstract please provide some background information of the proposed study.

Response 2: Thank you very much for this important comment. We have added background information in the abstract. (Lines 16-19)

Point 3: In all table legends, please mention the number of samples used for calculating the values.

Response 3: Thank you very much for this important comment. We have added the number of samples in all table legends. (Lines 223, 252, 292, and 311)

Point 4: There are several typographical, and errors throughout the manuscript.

Response 4: Thank you very much for pointing this out. We have made the revision in the manuscript.

We appreciate for your warm work earnestly and hope that the correction will meet with approval.

Special thanks to you for your good comments.

Best regards

Sincerely yours,

Jinyu Tian and co-authors

Jiangsu Key Laboratory of Crop Cultivation and Physiology / Innovation Center of Rice Cultivation Technology in Yangtze Valley, Ministry of Agriculture / Co-Innovation Center for Modern Production Technology of Grain Crops, Yangzhou University, Yangzhou 225009, China

Round 2

Reviewer 1 Report

The revised manuscript seems to be properly amended.

Reviewer 2 Report

Authors have responded satisfactorily to my comments. The manuscript in its current form may be accepted for publication.